# Antimicrobial Functionalization of Prolamine–Silica Hybrid Coatings with Fumaric Acid for Food Packaging Materials and Their Biocompatibility

**DOI:** 10.3390/antibiotics11091259

**Published:** 2022-09-16

**Authors:** Franziska Trodtfeld, Tina Tölke, Cornelia Wiegand

**Affiliations:** 1Department of Dermatology, University Medical Center Jena, Erfurter Str. 35, D-07740 Jena, Germany; 2INNOVENT e.V., Prüssingstraße 27 B, D-07745 Jena, Germany

**Keywords:** antimicrobial activity, barrier coating, hydrogel, food packaging, material–cell interaction, biocompatibility

## Abstract

The interest of the food packaging industry in biodegradable, recyclable, and functional materials has steadily increased in recent years. The use of hydrogels in the food sector holds great potential for use in packaging systems or as carriers for bioactive substances. The synthesis of an oxygen barrier coating of prolaminic silica material and antimicrobial functionalization with fumaric acid for packaging materials described here is an elegant way to meet these requirements. The developed material achieved a significant antimicrobial activity against *Escherichia coli* and *Staphylococcus aureus*, two common clinical pathogens. Another pre-requisite of such materials is a high biocompatibility, which can be assessed using human cell models, to help ensure consumer safety. The biocompatibility was determined by luminescence adenosine triphosphate and photometric lactate dehydrogenase assays. No cytotoxic effects on human keratinocytes in vitro were found for the test materials.

## 1. Introduction

The accumulation of synthetic plastics and the environmental problems they have generated have resulted in an increased search for alternative coatings and materials based on natural polymers such as polysaccharides and proteins [1,2]. Food packaging materials are developed to ensure or prolong the shelf life of the packaged product. Food items are prone to the oxidation or contamination of the product. The oxidation process leads to spoilage in off-flavors and colors as well as texture alterations, whereas the contamination of the product leads to an increase in health risks to the consumer. The use of bio-based hydrogels for food packaging has increased in recent years [3,4,5], but their commercial use remains limited. They are very flexible, elastic, and biocompatible, which makes them well-suited for use as packaging materials. Hydrogels can absorb a lot of water; therefore, they can reduce the water activity in a packaged product and slow down the spoilage of packaged goods [6]. Hydrogels are used to preserve hygroscopic foods such as powders or to maintain the texture of dry and crisp products. Various materials and methods have been investigated to expand the applications of hydrogels such as incorporating active materials into the porous structure of the gel, improving the absorption properties, and using them as biosensors to ensure food freshness [7,8,9,10,11]. Although there are already studies on extending the shelf life of foods with hydrogels made from natural polymers by controlling the moisture, this approach of an antimicrobial coating with an oxygen barrier based on a hydrogel is new and promising because the properties of these materials can add significant value to packaged products due to their biocompatibility and low toxicity [12].

Paper is a recyclable and biodegradable packaging material, which is produced from renewable resources and utilized in many food packaging applications. However, it is permeable for vapors and oxygen as well as moisture-sensitive, which limits its application potential in the industry, because food products are not protected from oxidation and degradation. Hence, commercially available paper packaging is often coated with synthetic materials such as polyethylene (PE), polyvinylidene chloride (PVDC), ethylene vinyl alcohol (EVOH), or acrylic coating [13]. These solutions are suboptimal for recycling processes and the resulting packaging poses an environmental burden. Consequently, recyclable and biodegradable coatings with a functional oxygen barrier and antimicrobial functionalization are needed.

Prolamine, the storage protein fraction of wheat gluten, is biodegradable and known for its oxygen barrier properties [14]. The sol–gel technology is a useful tool to produce surface coatings combined with prolamine, because it enables controlling the surface structure and main characteristics of a layer coating [15,16]. The amino acids that form peptide bonds are the basis of the forming network. The combination of the organic compounds and inorganic silica increases the flexibility and viscosity of the material and expands the potential applications to materials with a rough surface [17]. This is important when a material like paper is used, which has wide pores and a rough surface. To increase the oxygen barrier and reduce oxidation processes of the packaged goods, a thick coating for a paper product is needed due to its porous structure. To further prevent the contamination and spoilage of food products, which is important for lowering food waste and reducing consumer health risks, antimicrobial active substances can be included. Fumaric acid is widely used as an additive in the food industry and has also shown antimicrobial activity against multiple microorganisms [18,19]. SiO_2_ particles can be easily loaded with fumaric acid and encapsulated with a polyelectrolyte complex before being added to the surface layer to control the release of fumaric acid.

In this study, an innovative prolamine–silica hybrid coating of a paper material was established. For antimicrobial functionalization, the SiO_2_ particles were loaded with fumaric acid. A pH dependent release of the fumaric acid was ensured by encapsulation with a polyelectrolyte complex. The antimicrobial activity and biocompatibility of this paper coating was investigated as a prerequisite for consumer safety.

## 2. Results

### 2.1. Material Analysis and Characterization

In Figure 1, the material multilayer structure is outlined on the rough paper surface as well as the main chemical bonds responsible for the adhesion of the barrier prolamine–silica layer (ProMa) and the inorganic silica layer trimethylethoxysilane (MT) on top as well as the incorporated fumaric acid loaded particles (F). Figure 2 shows the SEM images of the prolamine–silica (ProMa) coating from the surface perspective and profile perspective for an understanding of the surface and internal coating structure. In the SEM images, microdefects due to brittleness of the coating could be observed [20].

According to Figure 3, the FTIR spectra (IR MB 3000, ABB Automation products GmbH, Alzenau, Germany) of the prolamine–silica hybrid coating (ProMa) and the silica sol synthesized from tetraethoxysilane showed the significant Si-O-Si peak of a silica sol (1072 cm^−1^) [21].

The absorbance peak at 3300 cm^−1^ present in all samples was attributed to the O–H stretching, the peak at 2948 cm^−1^ was accredited to C–H stretching, and the peak at 1405–1409 cm^− 1^ was assigned to the COO− stretching vibrations. Crosslinking with citric acid led to the carbonyl stretch peak at 1700 cm^−1^, which indicates the formation of ester groups (C=O) [22,23], the C–C–O stretch peak around 1200 cm^−1^, and the O–C–C stretch peak at 1100 cm^−1^ supports that the citric acid esterifies with –OH groups of D-mannose and the amino acids prolin and glutamin [24,25]. The silica sol does not esterify with the organic compounds; the Si–OH (956 cm^−1^) peak [26] from the hydrolyzed tetraethoxysilane based sol was still present in the ProMa coating, which suggests that the silica did not react with the organic compounds and was distributed as silica particles in the film. Subsequent to this analysis, the phase identification was determined (AFM, MFP 3D-Classic, Asylum research) to detect how the inorganic and organic components were arranged in the layer. As Figure 4 indicates, both phases were evenly distributed and formed a homogenous system.

The prolamine–silica coating was not water resistant due to only light crosslinking and the still existing hydroxyl groups on the surface. This was intentional as the polar hydroxyl groups and the amino groups in this material are mainly responsible for the oxygen barrier (Table 1) [27]. The solubility ensures the detaching of the coating from the cellulose fiber in the recycling process. The top layer, silica sol based on trimethylethoxysilane, was added to increase the contact angle up to 78° (OCA 15+; Data Physics) (Figure 5).

The material properties of the packaging material were investigated by determining the transmittance (%) of the coatings (Figure 6a) by UV–Vis spectra (Lambda 2, PerkinElmer). This was undertaken to investigate the extent to which the coating changed the transparency or, as in this case, the color of the base material. The stress–strain diagram (Figure 6b) acquired using Texture Analyzer TA. XT2i (Stable Micro Systems Ltd., Godalming, UK) demonstrates the degree to which the coatings affected the extensibility, strength, and brittleness of the paper.

Figure 6a shows that the hydrogel coating slightly improved the transmission of the glass. This has also been observed for other porous coatings [28,29]. However, after application of the inorganic topcoat, the transmittance deteriorated and dropped to 80%. The addition of the antimicrobial particles further decreased the transmittance between the wavelength of 700 and 400 nm, depending on the amount of added particles. The paper also became more elastic with the hydrogel coating but offered less load resistance, therefore the yield stress decreased (Figure 6b). The breaking strain did not change in response to the hydrogel coating but slightly decreased. However, the yield stress was found to be increased with the multilayer coating, but also increased the yield stress again. No deformation of the material was observed [30,31,32].

The oxygen permeability was measured to determine whether the material served its purpose as an effective food packaging material (Table 1).

The oxygen permeability for the barrier coating ProMa and ProMa MT showed very high deviations. This indicates that the microdefects caused by brittleness and swelling of the surface-bound gel increased the oxygen permeability. The application of silica sol as a top layer did not influence the barrier coating significantly.

To ensure the pH-dependent release of fumaric acid from the loaded silica particles, the release behavior at different pH values was analyzed to determine the release of fumaric acid depending on the pH of the extraction buffer (Figure 7).

The release of fumaric acid from the encapsulated SiO_2_ particles was analyzed by dispersing the particles in pH 3 and pH 5 buffer. The detected absorbance with UV–Vis spectra validated that the encapsulation of the particles led to a difference in the fumaric acid release in relation to the pH of the buffer. The results in Figure 7 show that at pH 3, the particles released a lower amount of fumaric acid; at pH 5, an increased acid release was observed. During the loading process, the particles were not stirred while drying. The release detection showed that the loading of the particles was successful as well as the encapsulation and the intended release behavior. 

### 2.2. Functionality and Biocompatibility Testing

The antimicrobial activity of the developed materials with antimicrobial functionalization was performed with *E. coli* DSM 5923 and *S. aureus* ATCC 6538. The test was carried out according to ISO 22196 and showed a strong decrease in microbial growth (log reduction > 3.0; Figure 8) [33]. Figure 8 shows the results for different amounts of loaded particles incorporated into silica sol to determine the minimum number of particles, which led to a significant reduction in microbial growth (log cfu) for the pathogen *E. coli* and *S. aureus.*

The suppression of microbial growth for *E. coli* DSM 5923 and *S. aureus* ATTC 6538 verified the successful antimicrobial functionalization of the material surface. A sample (PP MT F) without the barrier coating (ProMa) was tested to confirm that the antimicrobial effect was induced by the incorporated particles of the top layer and did not trace back to the barrier coating. The log reduction of 6.67 for the sample without the barrier coating demonstrates that the barrier coating was not needed for the effective inhibition of microbial growth. In Figure 8, the microbial growth decreased with an increase in the loaded particles in the material coating for *E. coli* whereas for *S. aureus*, the amount of 0.01 g of particles in 1 g silica sol was already enough for a strong log reduction (>3).

To determine the biocompatibility of the materials, the adenosine triphosphate and lactate dehydrogenase assays were performed using human keratinocytes as well as apoptosis with the Caspase-Glo 3/7 assay, which is a luminescent assay that measures caspase 3 and caspase 7 activity (Figure 9, Figure 10 and Figure 11). Cytotoxicity for keratinocytes occurs through a decrease in the proliferation ability or an increase in dead cells [34].

The cell count [%] indicates the anti-proliferative impact of GP ProMa. In comparison, GP ProMa MT did not show an anti-proliferative effect, which indicates that the trimethylethoxysilane top layer inhibits the dissolution of the coating effectively. The extract of GP ProMa MT F also exerted an anti-proliferative effect comparable to GP ProMa. The effect was not observed when DMEM was used instead of the defined sweat solution. Apoptosis was determined with the caspase-Glo 3/7 assay; for a positive control, actinomycin D was used. There was no apoptotic effect detected by analyzing the material sample treated cells. Direct cytotoxic effects of the materials causing membrane disruption, which is characteristic for cell necrosis, were determined by measurement of the LDH release from the cells (Figure 11).

Only marginal amounts of lactate dehydrogenase (LDH) were detected when cells were incubated with the test materials. In accordance, the calculation of cytotoxicity [%] from the data indicates no negative effects of the materials against human keratinocytes.

## 3. Discussion

The use of hydrogels in food packaging systems has grown rapidly in recent years, focusing mainly on moisture control [35,36,37]. The application of a hydrogel layer system for antimicrobial compound release and oxygen barrier in combination has not yet been investigated in detail. The selection of suitable polymers to produce the hydrogel ensures the manufacture of biodegradable packaging materials with a suitable degradation rate. This depends on several parameters such as the monomer, the crosslinker, and the crystallinity. The initial molecular weight is relevant, as mass degradation is only possible when the oligomer fragments are small enough to be water soluble [38]. The use of paper packaging in the food industry is increasing due to its great recyclability and waste stream management. Paper packaging materials are gas-permeable, so for sensitive packaged products, they need an additional coating layer or lamination that reduces the oxidation processes and contamination [39]. For food material application, synthetic coatings based on polyethylene (PE), polyvinylidene chloride (PVDC), polyethylene alcohol (PVA), ethylene vinyl alcohol (EVOH), or acrylic coatings are often used, which decrease the recyclability of the paper material [13,40,41]. Therefore, interest in recyclable and biodegradable functional coatings for paper packaging is growing. In addition to maintaining food quality, a functional gas barrier and an antimicrobial strategy are beneficial for consumer health. Pathogens responsible for relevant food-borne illnesses include *E. coli* and *S. aureus.* Some of the synthetic, commercially available antimicrobial agents used to control the growth of pathogens increase antibiotic resistance [42]. Therefore, interest in new effective ways to inhibit microbial growth is increasing. It is further essential that antimicrobial agents do not exert any cytotoxic effects on the consumer upon contact. This was, for instance, demonstrated for the antimicrobial agents ZnO or Ag, which can have a detectable high cytotoxic effect on human keratinocytes [43]. Multiple other studies have described the inhibition of food-borne pathogens by organic acids [44,45]. Organic acids feature good biocompatibility [23]. In addition, they might function as a more environmentally friendly solution as metallion-based antimicrobials and antibiotics released into the environment and water may lead to an increase in microbial resistance [42,46]. Fumaric acid is widely used as an acidulate in the food industry but has not been broadly applied as an antimicrobial agent yet.

This study describes the synthesis and characterization of a biodegradable and recyclable multilayer coating for food packaging materials based on a hydrogel from prolamine and silica sol. Prolamines have film forming properties and polar groups, which are of significance when creating a barrier for a unipolar molecule such as oxygen. The hybrid material showed good oxygen barrier properties of 61 [100 µm∙cm³/(m²dbar)], which was comparable to acrylate-based coatings for cartons, and would be applicable for dairy products, fresh products, or baked goods [47]. The crosslinking agent citric acid and D-mannose as well as the prolamines esterify and form a network structure. The peak at 1700 cm^−1^ was assigned to the forming ester group, which led to hydrogel formation [22,23,25]. The inorganic silica sol is distributed evenly in the organic network and optimizes the material surface and structure; the coating has a non-sticky surface and is easy to handle. Although the silica sol and organic compounds did not esterify, as indicated by the Si–OH peak in the prolamine–silica film, they still formed a homogenous material as suggested by the phase identification images. The dispersed silica particles interact with the organic network through ionic interactions and hydrogen bonding, crosslinking physically instead of covalently. Gaharwar et al. [48] described that the polymer chain can be absorbed and desorbed on charged silica surfaces, which leads to an increase in the viscoelastic capabilities [49,50,51]. This increases the stiffness and durability of the coating and leads to the mentioned improvement in the surface structure. At the same time, the crosslinking of the components is dynamic enough that the system remains water soluble, which is supported by the stress–strain curve (Figure 6b). The hydrogel coating increased the flexibility of the paper, while simultaneously decreasing the yield stress. The elongation at break was not significantly reduced, which means that the coating did not improve the ductility of the material [30,32]. The coating process leads to swelling of the cellulose fiber and allows the hydrogel coating to penetrate the paper, affecting the interactions between the fibers. Consequently, the coating led to lower fiber–fiber interaction and thus the lower tensile strength of the paper material [52,53]. The inorganic topcoat increased the stiffness of the material and led to a slight increase in yield stress [54]. The pure hydrogel coating has a very good light transmission, which is probably due to the very small pore size of the material [29]. The light transmission decreases when the inorganic top layer is applied, especially when the antimicrobial particles are added (Figure 6a). The particles are ~1 µm in size to protect consumers from potential nanoparticle migration into the food products or ingestion [55,56,57]. The lower light transmission is not particularly relevant for an opaque material such as glassine paper. However, if a transparent substrate such as glass or a transparent film is used, the particle size as well as the dispersion of the silica particles can be optimized to improve the light transmission. By modifying the particle surface, agglomeration can be reduced and thus light transmission can be improved [58]. The SEM images showed some microdefects due to brittleness in the coating that could occur during the drying process, which is critical when working with amino acids. Hydrogels are polymers that absorb water effectively; absorption allows a surface-bound gel to swell only unidirectionally instead of in all directions. This leads to biaxial compressive stress in the gel, which is expressed in the folding of the gel [59,60]. These microdefects were mainly responsible for the high standard deviation observed for the oxygen permeability of the product [20]. Further investigations should include the addition of plasticizers, which are more effective in reducing brittleness and renders the coating even more flexible [61]. They could also improve the water vapor barrier of the top layer to reduce rehydrating of the barrier coating and therefore inhibit the creasing process. The water solubility of the barrier coating ensures separation of the coating from the cellulosic fiber, which is important in order not to reduce the number of recyclable cycles the fiber undergoes. In addition, the coating can be disposed of with the water waste stream and does not contaminate screens, filters, or rolls, which would lead to production losses. At the same time, a water-soluble coating leads to a less stable material, which is not suitable for moist products or very humid climates. In accordance, a more hydrophobic top layer was introduced that bonds at the free hydroxyl groups on the barrier coating surface and increases the surface contact angle. This led to an increase in water repellence and could expand the material application profile as well as prolong the storage life.

The material showed an effective antimicrobial functionalization of the surface against *E. coli* and *S. aureus* (log reduction > 3). The antimicrobial effectiveness of fumaric acid was also demonstrated by other studies. For example, lettuce and cabbage were directly treated with fumaric acid and a log reduction of 2.0 for *E. coli* was detected on the lettuce. On the cabbage, a successful biofilm eradication of *E. coli* and *S. aureus* with a log reduction of >3 was demonstrated [62,63]. For antimicrobial textiles, a significant antimicrobial activity is classified as a log reduction of ≥1 to <3 [33]. The minimal amount needed for a significant antimicrobial activity of fumaric acid loaded particles, which are incorporated into the top layer, for *E. coli* was an amount of 0.03 g of fumaric acid loaded particles, which exerted a log reduction of 2.47. However, the sample with 0.02 g of the fumaric acid loaded particles demonstrated a log reduction of 0.95. For *S. aureus,* samples with 0.01 g of particles already achieved a strong log reduction of >3, which shows that this material is highly effective against *S. aureus*. The entire mechanism of organic acid-induced antimicrobial activity is not yet understood. The assumption is that at an intracellular neutral pH, the organic acids dissociate into protons and anions. When the outside pH is lower than the intracellular pH, acid anions accumulate inside the cell. The presence of organic anions appears to be crucial for the inhibition of microbial growth, as shown by a comparison with HCl and several organic acids [64]. Even though both acids were effective in lowering the pH, the organic acids inhibited the microbial growth better. The antimicrobial functionalization of the material surface is intended to release the fumaric acid from the particles through contact with water. The applied layer-by-layer coating controls the release behavior; the polyelectrolyte complex formed in the acidic chitosan and pectin solutions traps the fumaric acid in place. At higher ambient pH, the carboxylic groups present become ionized, which induces static repulsion between ionized groups. This leads to an increase in H_2_O in the polyelectrolyte complex, resulting in the release of fumaric acid, which can diffuse out of the network [65,66]. The particles are not agitated during drying; the homogeneity of the products could be improved by, for example, spray drying. This could also lead to a reduction in the sample-to-sample variability. Contact of the loaded and capsuled particle with water will cause the fumaric acid to dissociate, lowering the pH, and the fumaric acid anions will accumulate in the cell and inhibit microbial growth. The log reduction of >3 for the materials against *E. coli* and *S. aureus* supports this theory. This antimicrobial activity is comparable to other antimicrobial agents such as silver additives and acyl pyrazolone [67]. Both samples, GP ProMa and GP ProMa MT F, decreased the pH of the sweat solution from pH 6 to pH 3 (data not shown). The transport of the excess protons to the extracellular environment requires cellular ATP, which may inhibit cellular function due to a decrease in cellular energy [63]. Decreased cellular ATP affects cell division and can lead to the obstruction of cell mitosis [68]. The hindrance during cell divisions leads to slower cell growth compared to the control, which results in less detectable ATP with the luminescence assay. This is a possible explanation for the anti-proliferative effect exhibited on human keratinocytes in contact with samples GP ProMa and GP ProMa MT F. Short chain carboxylic acids affect the dose-and acid dependent cell proliferation, apoptosis, and necrosis [69]. A decrease of 0.3–0.4 pH units lower than normal is known to trigger apoptosis and adjusting the cell culture medium to pH 5 with hydrochloric acid left only 10% of viable cells [70]. Therefore, a decrease in cell numbers due to the induction of apoptosis could be possible. The mitochondria of normal cells have an electrochemical gradient regarding the pH and volume components, primarily created by a net efflux of H^+^ ions from the matrix to the outer surface of the inner membrane. The decrease in pH could lead to a depolarization of mitochondria and the loss of the electrochemical gradient, which is associated with apoptosis [71]. Therefore, an apoptosis assay, which detects caspase 3 and caspase 7, was performed. No apoptosis was detected for the material samples used, which suggests that the samples only induce an anti-proliferative effect without eliciting programmed cell death. The LDH assay carried out showed no cytotoxic effects of the functionalized materials. The results classify the materials as consumer safe according to DIN EN ISO 10993-5.

In summary, a material was developed that exhibited strong antimicrobial activity against *E. coli* and *S. aureus* (log reduction > 3), comparable to commercially available products. Fumaric acid, which is already used as an acidifying agent in the food industry, was used as an additive. In addition, a hydrogel based on biodegradable components was synthesized and successfully provided with an oxygen barrier, suitable for fresh, baked, dairy, and meat products. For further research, a combination of an oxygen barrier and a superabsorbent material could be useful. The recycling properties were improved by ensuring the continuous use of the paper fiber through recovery and good separation of the coating. The material was tested with human keratinocytes for biocompatibility, and no cytotoxic or apoptotic effects were detected, ensuring consumer safety. Further research should address the possibility of allergenic potential and exclude it.

## 4. Materials and Methods

### 4.1. Synthesis Hybrid Coating 

For the prolamine extract, 20 g wheat gluten (>75% protein, Roth, Karlsruhe, Germany, 100 g) was extracted with 200 mL of 70 % ethanol/H_2_O dest solution. The solution was centrifuged for 10 min by 4000 rpm. Then, 10 mL of the supernatant was stirred with 0.5 g D-mannose (Sigma-Aldrich, St. Louis, MO, USA, 100 g). Tetraethyl orthosilicate (Sigma-Aldrich, 500 mL) was hydrolyzed beforehand with citric acid (VWR, BDH chemicals, Radnor, PA, USA, 1 kg), 3 mL was added to the solution. Then, 1.2 g of citric acid was added, and the solution was stirred for 24 h at 300 rpm. Trimethylethoxysilane (abcr, Karlsruhe, Germany, 1 L) was hydrolyzed and stirred for 1 h at 300 rpm. The paper was coated with 400 µm wet film of the barrier coating and dried at 80 °C in an oven for 30 min. The loaded particles were dispersed in the silica-sol based on trimethylethoxysilane, applied in wet film onto the barrier coating, and dried for 1 h at 80 °C.

### 4.2. Synthesis and Loading of SiO_2_ Particles

The SiO_2_ particles were synthesized following the description of Pérez-Esteve et al. (2016) [72]. The particles were not autoclaved but stored in an oven at 120 °C for 24 h before they were placed in the muffle furnace. Afterward, 5 g of SiO_2_ particles were dispersed in an aqueous fumaric acid (Sigma Aldrich, 100 g) solution (4.9 g/L) and stirred for 24 h. After that, the solution was dried in an oven at 100 °C until all of the water evaporated. The loaded particles were dispersed in chitosan solution (1 mg/mL; 0.5 M NaCl, pH 3) (Sigma Aldrich, 100 g, low viscosity) and stirred at 300 rpm for 1 h. Next, the particles were vacuum filtered and dried at 80 °C in the oven. The particles were then dispersed in pectin solution (1 mg/mL: 0.5 M NaCl, pH 3) (Sigma Aldrich, 100 g) for 1 h and dried at 80 °C in the oven. These coating steps were alternated and repeated twice.

### 4.3. Material Properties

SEM imaging was performed using the Supra 55 VP (ZEISS AG, Oberkochen, Germany) with EDX system Quantax (Bruker Corporation, Billerica, MA, USA).

Tensile strength is defined as the breaking strength of a material under stress. From the measured stress–strain curve, the strength, plasticity, brittleness, or elasticity of a material can be determined. The stress–strain curve was determined using the TA.XT2i texture analyzer (Stable Micro Systems Ltd., Godalming, UK). Specimens were cut into 0.5 cm × 3 cm rectangular strips. The initial grip spacing was set at 20 mm. The test speed at which the specimens were pulled was 1.00 mm/s. The test was stopped when the material broke. Six samples were used to determine the strain–stress curve and the mean values are reported.

The transmittance of the coatings was determined by UV–Vis spectroscopy. The coatings were applied to super-premium microscope slides (ground edges 90°, 1.0–1.2 mm, VWR). The ProMa coating was applied with a wet film of 10 µm and the inorganic silica coating with a wet film of 4 µm. The samples were dried in a drying oven and then the transmittance was measured in duplicates at a wavelength of 800 nm to 300 nm in a UV–Vis spectrometer (Lambda 2, PerkinElmer, Waltham, MA, USA).

### 4.4. Release Behavior

The loaded and capsuled particles (40 mg) were dispersed in 5 mL of buffered solution added into a dialysis tube (MWCO 3000–6000) and added into 1 L of the same buffer solution (pH 3, pH 5). The solution was stirred at 100 rpm, 3 mL samples were regularly taken during the time period and the volume was refilled with the related buffer solution. The samples were measured with UV–Vis spectroscopy (300–190 nm) (Lambda 2, PerkinElmer). The concentration was calculated with a calibration curve of fumaric acid, and the concentration values were corrected using Equation (1) [73], where *c_n_* is the concentration of the solution, *V_s_* is the volume of the sample, *V_t_* is the total volume of dissolution medium, and *c_i_* is the concentration at the *i*th time point.
(1)$${\mathrm{conc}}_{\mathrm{corr}}=c_{n}+\frac{V_{s}}{V_{t}}\cdot \;\overset{n-1}{\underset{i=1}{\sum }}c_{i}$$

For the pH 3 buffer solution, citric acid, sodium hydroxide solution, and sodium chloride were used; for the pH 5 buffer, only citric acid and sodium hydroxide solution were used.

### 4.5. Barrier Function

The oxygen permeability of the samples was determined using the gas permeability tester GTT and GDP-C (Brugger Feinmechanik GmbH, Munich, Germany). Oxygen was used as the test gas; samples were masked on both sides, leaving a test area of 10 cm^2^. The chamber was tempered at 23 °C and 17% RH. The oxygen flow was 0.5 mL/min.

### 4.6. Testing of Antimicrobial Activity

The samples were tested in accordance with ISO 22196. The sample surfaces (1 × 1 cm^2^) were sterilized using a 1 min UV-light (Inno-Line; Jupiter Kitchen Tools GmbH) treatment. The microorganisms *E. coli* DSM 5923 and *S. aureus* ATCC 6538 were obtained from the DSZM. Tests were performed in triplicate. Polypropylene foil was used as a control. All samples were inoculated with bacteria suspension and incubated for 24 h at 37 °C. Isotonic saline solution with Tween 20 was used to recover viable bacteria from the sample surfaces. Serial dilutions from the bacteria suspensions were prepared and plated on Columbia agar with 5% sheep blood (bioMérieux SA, Marcy-l’Étoile, France). The plates were incubated for 24 h at 37 °C. The antimicrobial effect was calculated using the following equation:(2)$$R=\left(U_{t}-U_{o}\right)-\left(A_{t}-U_{o}\right)=U_{t}-A_{t}$$

### 4.7. Testing for Biocompatibility

The cell compatibility was determined by fabricating sample extracts following DIN EN ISO 10993-12. Therefore, 200 mg of the material samples was mixed with 10 mL of Dulbecco’s modified Eagle medium (BioConcept Ltd., Allschwill, Switzerland) or 10 mL-controlled sweat solution (5 g NaCl; 2.2 g NaH_2_PO_4_∙2H_2_O; 0.5 g C_6_H_9_O_2_N_3_∙H_2_O + HCl; 0.1 N NaOH—pH 5.5) in Erlenmeyer flasks and shaken for 24 h at 37 °C in a ThermoBath (GFL, Deutschland, Burgwedel, Germany). Subsequently, the material residues were removed, and extracts were sterilized by filtration (0.2 µm filter, Sarstedt AG & Co. KG, Nümbrecht, Germany). Cells were cultivated in DMEM with 1% antibiotic-antimycotic solution (10,000 U/mL penicillin, 10,000 µg/mL streptomycin, 25 µg/mL amphotericin; PromoCell, Heidelberg, Germany) and 10% fetal calf serum (PAN Biotech, Aidenbach, Germany) for seven days in 75 cm^2^ cell culture flasks at 37 °C in a humidified atmosphere with 5% CO_2_. Cells were detached with trypsin ethylenediaminetetraacetic acid (Gibco, Life Technologies Limited, Carlsbad, CA, USA) sown into 96-well plates at a cell density of, 5 × 10^4^/cells/mL and incubated for 48 h. Afterward, the medium was exchanged for fresh medium or controlled sweat solution (negative control), material extracts, and Triton-X 100 (cytotoxicity control), followed by incubation for 1 h, 24 h, and 48 h. The viability and reproductivity of the cells were detected using the ATPLite M Assay (PerkinElmer, Waltham; MA, USA) according to the manufacturer’s instructions; apoptosis was detected using the caspase-glo 3/7 assay (Promega Corporation, Madison, WI, USA) following the manufacturer’s instructions. The cell count was calculated as [%] of the negative control at the respective time point. The material cytotoxic effects were determined using the cytotoxicity detection kit (Roche, Basel, Switzerland) according to the manufacturer’s instructions. Optical density was measured at 490 nm (SPECTROstar-Omega, BGM Labtech, Ortenberg, Germany). Cytotoxicity was calculated using Equation (3).
(3)$$\mathrm{cytotoxicity}\;\left[\%\right]=\frac{n-\mathrm{cytotoxicity}\;\mathrm{control}}{\mathrm{negative}\;\mathrm{control}-\mathrm{cytotoxicity}\;\mathrm{control}\;\times 100}$$

### 4.8. Statistical Analysis

SSPS was used to perform a Mann–Whitney-U-Test to determine the central tendencies of two independent random samples. Asterisks indicate significant deviations from the negative control (* *p* < 0.05; ** *p* < 0.01; *** *p* < 0.001).

## 5. Conclusions

In this study, it was shown that the field of application of hydrogels can be extended to the oxygen barrier for food packaging. Furthermore, effective antimicrobial functionalization of the packaging material by pH-dependent release of the active ingredient was demonstrated, and biocompatibility for the end consumer was proven by the selected assays. The prolamine–silica film is comparable in performance to commercial coating formulations while providing a renewable, recyclable, and biodegradable consumer safe alternative.

## Figures and Tables

**Figure 1 antibiotics-11-01259-f001:**
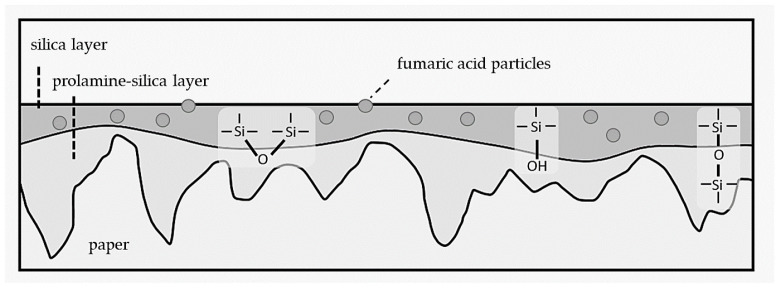
The profile scheme of the applied multilayer on the paper substrate.

**Figure 2 antibiotics-11-01259-f002:**
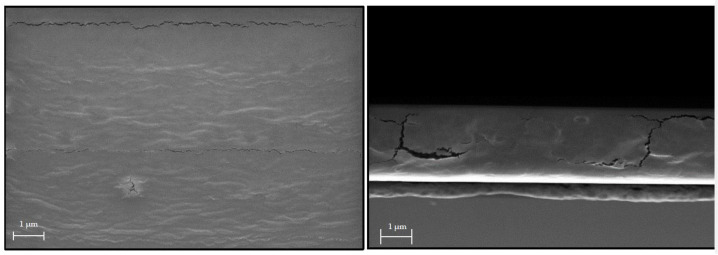
SEM images of the ProMa coating on a silica wafer (2 cm × 2 cm) from the profile and surface view; samples were coated with ProMa and dried for 2 h by 80 °C (surface gold sputtered).

**Figure 3 antibiotics-11-01259-f003:**
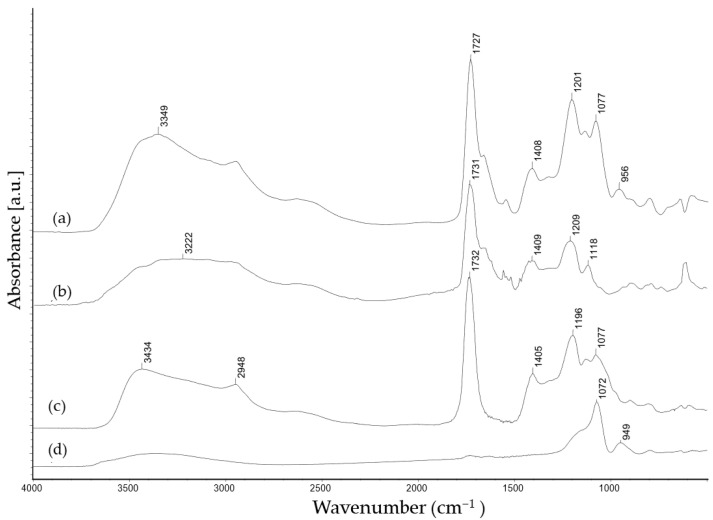
The FTIR transmittance of films on a silica wafer cm^−1^: (**a**) ProMa coating, (**b**) reaction product D-Mannose and citric acid (**c**) reaction product of prolamines and citric acid, (**d**) silica sol based on tetraethoxysilane hydrolyzed with citric acid.

**Figure 4 antibiotics-11-01259-f004:**
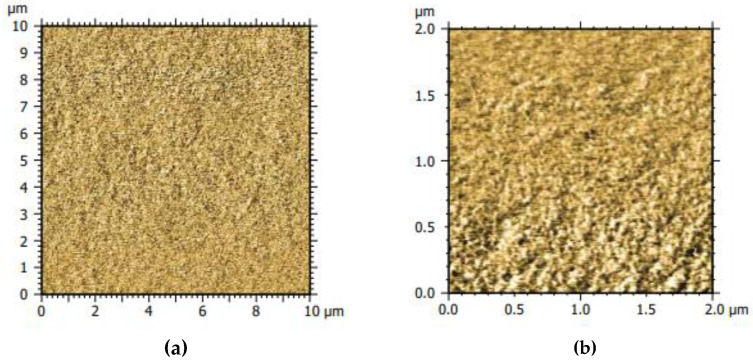
The AFM phase identification pictures of the ProMa coating surface on silica wafer (**a**) section 10 µm × 10 µm, (**b**) section 2 µm × 2 µm.

**Figure 5 antibiotics-11-01259-f005:**
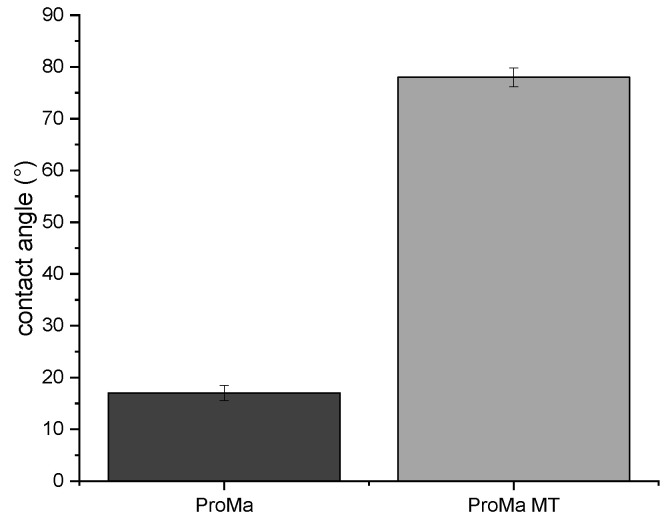
The contact angle [°] of the ProMa coating and ProMa + trimethylethoxysilane on the glass (5 cm × 5 cm) coated and dried by 80 °C for 2 h.

**Figure 6 antibiotics-11-01259-f006:**
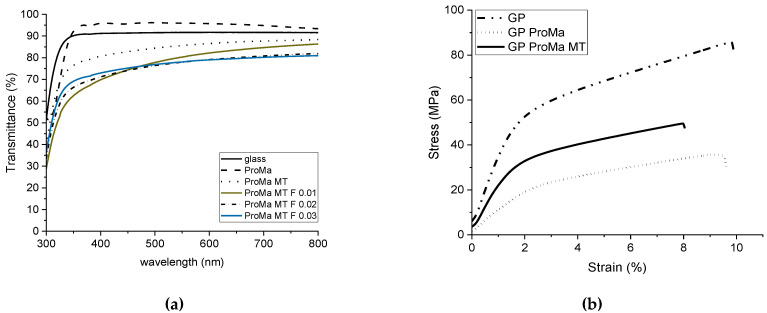
(**a**) UV light transmission of glass and the coatings ProMa, ProMa MT, ProMa MT F (0.01; 0.02; 0.03) on glass. (**b**) Stress (mPa) plotted against the strain (%) of glassine paper and glassine paper coated with ProMa and ProMa MT.

**Figure 7 antibiotics-11-01259-f007:**
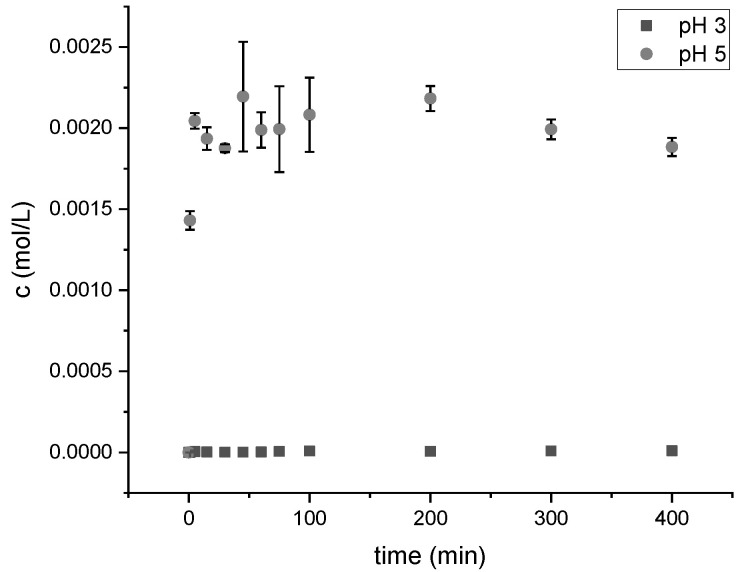
The release curve of the 40 mg loaded silica particles in buffer pH 5 and pH 3, UV–Vis spectra (Lambda 2, PerkinElmer) wavelength 300–190 (nm), *n* = 3 (232 nm pH 5, 230 nm pH 3).

**Figure 8 antibiotics-11-01259-f008:**
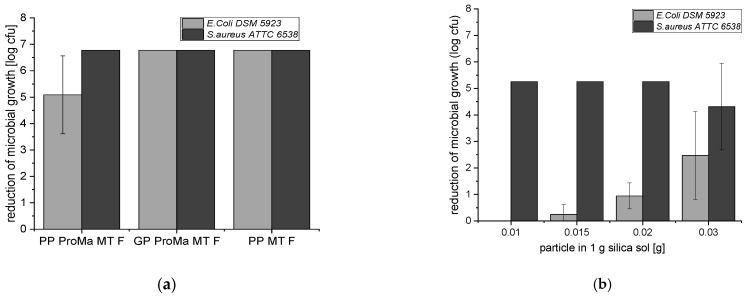
(**a**) Germ reduction in [log cfu] for *E. coli* DSM 5923 and *S. aureus* ATTC 6538 on sterilized substrate surfaces (PP: polypropylene, ProMa: prolamine–silica sol coating, MT: silica sol (trimethylethoxysilane); F: fumaric acid particles; GP: glassine paper 50 g/m^2^); (**b**) germ reduction in [log cfu] for *E. coli* DSM 5923 and *S. aureus* ATTC 6538 on the sterilized substrate surfaces.

**Figure 9 antibiotics-11-01259-f009:**
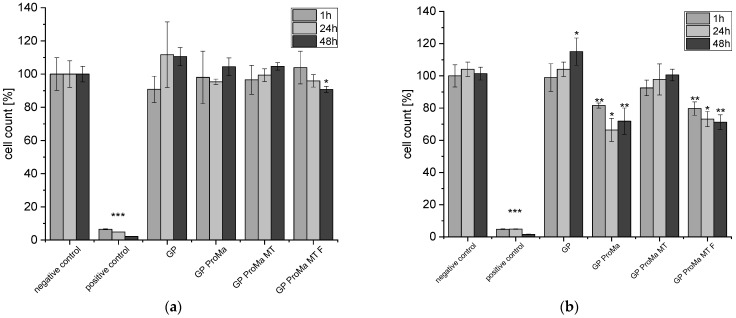
(**a**) Cell count [%] of the HaCaT cells (human keratinocyte cell line) exposed to the sample extracts in DMEM for 1 h, 24 h, 48 h; (**b**) cell count [%] of the HaCaT cells exposed to the sample extracts in defined sweat solution for 1 h, 24 h, 48 h. Asterisks indicate significant deviations from the negative control (* *p* < 0.05; ** *p* < 0.01; *** *p* < 0.001) (ProMa: prolamine–silica sol coating, MT: silica sol (trimethylethoxysilane); F: fumaric acid particles; GP: glassine paper 50 g/m^2^).

**Figure 10 antibiotics-11-01259-f010:**
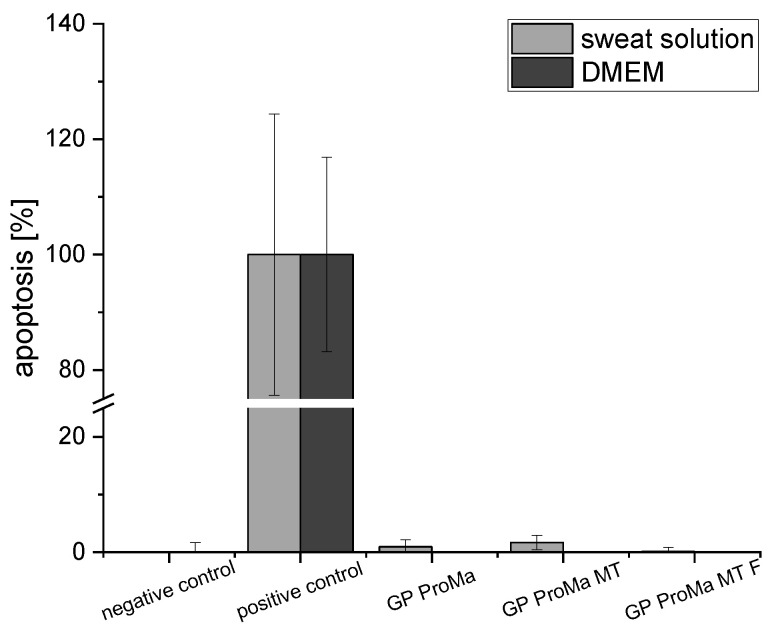
Apoptosis [%] of the HaCaT cells exposed to the sample extracts in DMEM and sweat solution for 24 h. For the positive control, actinomycin D was used.

**Figure 11 antibiotics-11-01259-f011:**
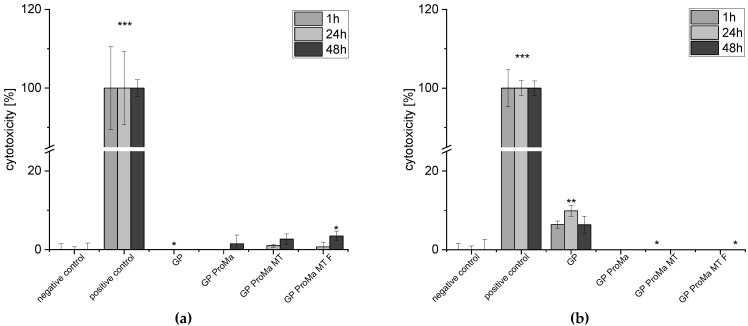
(**a**) Cytotoxicity [%] of the HaCaT cells exposed to the sample extracts in DMEM for 1 h, 24 h, 48 h; (**b**) cell count [%] of the HaCaT cells exposed to the sample extracts in defined sweat solution for 1 h, 24 h, 48 h. Asterisks indicate significant deviations from the negative control (* *p* < 0.05; ** *p* < 0.01; *** *p* < 0.001).

**Table 1 antibiotics-11-01259-t001:** The oxygen permeability (OP) of the ProMa barrier coating and the multilayer coating ProMa MT, *n* = 6.

Sample	ProMa	ProMa MT
OP [100 µm∙cm³/(m²dbar)]	82 +/− 87	61 +/− 70

## Data Availability

Not applicable.

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
