# Peer review of "Antimicrobial Functionalization of Prolamine–Silica Hybrid Coatings with Fumaric Acid for Food Packaging Materials and Their Biocompatibility"

_antibiotics, 2022, doi:10.3390/antibiotics11091259_

Round 1
Reviewer 1 Report
This manuscript describes the antimicrobial activity of prolamine-silica hybrid coatings with fumaric acid and applications in food packaging industry. The experiment design and presentation of results are good. Though minor editing of the text (e.g some sentences are starting with numbers in methodology), the manuscript is presented clearly without any major errors
Author Response
Response to Reviewer 1
Review 1
This manuscript describes the antimicrobial activity of prolamine-silica hybrid coatings with fumaric acid and applications in food packaging industry. The experiment design and presentation of results are good. Though minor editing of the text (e.g some sentences are starting with numbers in methodology), the manuscript is presented clearly without any major errors.
Thank you very much for your review.
The article has been carefully checked and the grammatical and punctuation errors have been corrected as suggested.
Reviewer 2 Report
The current manuscript describes the development and antimicrobial fumaric acid-functionalized prolamine-silica hybrid coatings for food packaging materials and the investigation of their biocompatibility. I think it is very interesting and can be published after a minor revision.
1- Figure 1: The designed Scheme is very vague, while it should represent the current research objectives. Has the UV light transmittance test been performed for the designed nanocomposite film?
2- The FTIR spectra shown in Figure 3 do not have the quality and resolution required to be used in a scientific paper. The authors should mark all the peaks and wavenumbers discussed in the main text and explain in more detail the results obtained and compare them with similar results in the literature and provide further references. Why did the author label the y-axis with extinction? It should be replaced by Transmittance%.
3- Line: 106-107 Figure 6 caption in Release profile: The author stated UV-vis spectra (Lambda 2, Perkin Elmer) wavenumber 300-190 nm…., Please replace wavenumber with wavelength (nm).
4- The introduction lacks a strong background study. The potential findings of other potential research must be included in the background.
5- Aren’t physical properties such as tensile strength and Young’s modulus significant factors in food packaging? If yes, have the films/hybrid coatings been investigated for the mentioned properties?
6- The author needs to check this manuscript for English proofreading, as I found some grammatical and punctuation errors while reviewing the article.
Reviewer 3 Report
This article described the development of coating material sing prolamine-silica material and an antimicrobial functionalization with fumaric acid that can be applied for packaging material. The article was well-written. It can be published in this journal after the major revision as follows
In line 60, It seems the sentence was incoherent with the previous sentences. The Authors mentioned, “microdefects which were not mentioned previously. In 24, “microdefects” was written differently “micro-defects”.
Figures 7a and 7b seem to have similar data, but it was presented differently. It is also the same for Figure 8. I suggest showing only one of them. Another one can be located in the supplementary material or appendix. I read the discussion the author mostly discussed the reduction.
In lines 95-97, the sentence is too long and unclear and it is difficult to understand the sentence. It is better to separate to improve clarity and understanding. Was the word “it” correct? or is the correct one “its”?
Figures 10 and 11 need to be improved. There is a big difference between positive control and others. Since the percentage of cytotoxicity of positive control is much higher than in other samples, I suggested editing the figures where other samples with a much lower percentage of cytotoxicity can also be seen clearly.
The authors mentioned the recyclable material and recyclability of materials as the main interest of paper packaging in the food industry. However, in this study, no measurement showed the recyclability of the developed material. In the discussion, recyclability was mentioned but was still not related to the measurement. Therefore, it is suggested for the authors discuss recyclability according to the parameters investigated in this study.
In line 136, the author mentioned " Ci is the concentration at the ith time point”, but no Ci was written in Equation (1).
In 313-315, what do GTT and 313 GDP-C stand for?
The conclusion did not well cover the most important results. It was too general, and It needs to be more elaborated.
Abbreviations should be extended at the first mention.
Author Response
Response to Reviewer 3
Review 3
This article described the development of coating material sing prolamine-silica material and an antimicrobial functionalization with fumaric acid that can be applied for packaging material. The article was well-written. It can be published in this journal after the major revision as follows
Thank you very much.
- In line 60, It seems the sentence was incoherent with the previous sentences. The Authors mentioned, “microdefects which were not mentioned previously. In 24, “microdefects” was written differently “micro-defects”.
Thank you for bringing this to our attention. We have amended the paragraph accordingly. It now reads:
line 78: “In the SEM pictures, microdefects due to brittleness of the coating can be observed”.
- Figures 7a and 7b seem to have similar data, but it was presented differently. It is also the same for Figure 8. I suggest showing only one of them. Another one can be located in the supplementary material or appendix. I read the discussion the author mostly discussed the reduction.
This was amended accordingly. Figure 7a and 7b now only show the reduction of bacteria.
- In lines 95-97, the sentence is too long and unclear and it is difficult to understand the sentence. It is better to separate to improve clarity and understanding. Was the word “it” correct? or is the correct one “its”?
Thank you for pointing this out, we have restructured the text section to make it more understandable and it now reads:
line 138 The oxygen permeability was measured to determine whether the material served its purpose as an effective food packaging (table 1).
line 146: To ensure the pH-dependent release of fumaric acid from the loaded silica particles, the release behavior at different pH values was analyzed to determine a release of fumaric acid depending on the pH of the extraction buffer (figure 7).
- Figures 10 and 11 need to be improved. There is a big difference between positive control and others. Since the percentage of cytotoxicity of positive control is much higher than in other samples, I suggested editing the figures where other samples with a much lower percentage of cytotoxicity can also be seen clearly.
Thank you for this suggestion, we have updated the numbers mentioned and added breaks on the y-axis to improve visibility of all bars shown.
- The authors mentioned the recyclable material and recyclability of materials as the main interest of paper packaging in the food industry. However, in this study, no measurement showed the recyclability of the developed material. In the discussion, recyclability was mentioned but was still not related to the measurement. Therefore, it is suggested for the authors discuss recyclability according to the parameters investigated in this study.
We concur with the reviewer`s assessment. Accordingly, we have updated the text to discuss recyclability more fully in the context of material selection and synthesis components, as well as the circular economy of paper and the environmental impact of antibiotic resistance-promoting antimicrobials.
- In line 136, the author mentioned " Ci is the concentration at the ith time point”, but no Ci was written in Equation (1).
Thank you for pointing this out, we amended this accordingly and corrected the label Cr to Ci.
- In 313-315, what do GTT and 313 GDP-C stand for?
GTT (Gas Transmission Tester) and GDP-C (Gasdurchlässigkeitsprügferät) are the model descriptions of the gas permeability tester of the company Brugger Feinmechanik.
- The conclusion did not well cover the most important results. It was too general, and it needs to be more elaborated.
We updated the conclusion as suggested. The conclusion now reads:
In this study it was shown that the field of application of hydrogels can be extended to oxygen barrier for food packaging. Furthermore, effective antimicrobial functionalization of the packaging material by pH-dependent release of the active ingredient was demonstrated, and biocompatibility for the end consumer was proven by the selected assays. The prolamine-silica film is comparable in performance to commercial coating formulations while providing a renewable, recyclable, biodegradable consumer safe alternative.
- Abbreviations should be extended at the first mention.
Thank you for pointing this out. The abbreviations are now spelled out when first used.
Round 2
Reviewer 3 Report
Well-improved